# Efficient LLM Collaboration via Planning

## Abstract

Recently, large language models (LLMs) have demonstrated strong performance, ranging from simple to complex tasks. However, while large models achieve remarkable results across diverse tasks, they often incur substantial monetary inference cost, making frequent use impractical for many applications. In contrast, small models are often freely available and easy to deploy locally, but their performance on complex tasks remains limited. This trade-off raises a natural question: how can small and large models efficiently collaborate to combine their complementary strengths? To bridge this trade-off, we propose COPE, a test-time collaboration framework. A planner model first generates a plan that serves as a lightweight intermediate that guides a downstream executor model. Small and large models take turns acting as planner and executor, exchanging plans in a multi-stage cascade to collaboratively solve tasks. Through comprehensive experiments on benchmarks spanning mathematical reasoning, code generation, open-ended generation, and sequential decision-making, we demonstrate that COPE achieves performance comparable to large proprietary models, while substantially reducing the inference API cost. These results highlight planning as an effective prior for cost-efficient inference.

## 1 Introduction

Large language models (LLMs) have achieved impressive performance on complex tasks (Guo et al., 2025; Comanici et al., 2025). However, deploying them at scale presents a critical trade-off between model capability and computational cost. As larger models continue to improve in capability, they also grow increasingly costly to run (Achiam et al., 2023; Schick et al., 2023). Smaller models, in contrast, are far more deployable, but their limited capacity hinders them on complex tasks (Kojima et al., 2022; Wei et al., 2022).

This trade-off is especially clear in realistic deployment scenarios. Users typically interact with language models via mobile or edge devices, which cannot host large models (Leviathan et al., 2023; Chowdhery et al., 2023). Instead, large models often reside in the cloud and incur substantial inference costs. In this setting, relying solely on large models for every request is practically unsustainable. These constraints call for inference strategies that use smaller, lower-cost models whenever possible and selectively leverage larger, higher-cost models only when necessary. We focus on this practically important setting, as studied in prior work on cost-aware inference (Yue et al., 2023; Kolawole et al., 2024).

There is growing interest in cost-aware inference with small and large LLMs (Liu et al., 2025; Hsu et al., 2024). Existing methods typically operate in a multi-stage cascade manner: a small model first attempts the task, and if its output fails to satisfy a confidence-based criterion (e.g., a majority-voting threshold), the task escalates to a large model in the next stage (Yue et al., 2023; Kolawole et al., 2024; Chen et al., 2023). However, these approaches remain a form of independent delegation, as only one model operates in each stage rather than collaborating within a stage (Yue et al., 2023; Kolawole et al., 2024; Chen et al., 2023). This limits their ability to jointly perform complex tasks in a structured and interactive manner.

In contrast, we propose to structure this collaboration through *planning*. Instead of directly performing a task, a model first produces a high-level abstraction of the task (such as a goal of the task or a guideline to solve the task), which we call a plan. A plan guides downstream execution by another model. This planning step serves as a lightweight, transferable intermediate that allows models to scaffold each other's thinking. Thus, we explore the following research question:

Table 1: Accuracy (%) of model combinations on the MATH-500 dataset. 'Base' shows accuracy without a planner; other columns use GPT-mini, Llama-3B, or Llama-1B as planners.

| Executor | Base | GPT-mini | Llama-3B | Llama-1B |
|---|---|---|---|---|
| GPT-mini | 73.8 | 76.2 | 70.6 | 69.6 |
| Llama-3B | 42.8 | 53.0 | 37.6 | 32.8 |
| Llama-1B | 25.2 | 36.4 | 26.0 | 23.2 |

Table 2: Accuracy of Llama-1B with different plan types on the MATH-500 dataset.

| Plan Type | Accuracy (%) |
|---|---|
| None | 25.2 |
| Guideline | 23.2 |
| Goal | **30.2** |

---

**Research Question**

Can planning function as a mediator of collaboration between small and large models for efficient inference?

---

To answer this question, we introduce COPE (Collaborative Planning and Execution), a test-time inference framework that uses planning as a lightweight interface for cross-model collaboration. Unlike prior cost-aware cascades, which escalate by handing the entire task to a larger model, COPE allows a larger model to guide a smaller model's execution before full takeover. The process begins with a free model attempting both planning and execution. If the model lacks confidence in its output, planning and then execution are gradually escalated to a costly model. This structure allows models to collaborate adaptively: easy tasks are handled by free models, and harder tasks are handled by costly models.

COPE significantly reduces inference cost while maintaining performance comparable to the larger model. For example, on the MATH-500 dataset, COPE achieves 75.8% accuracy (higher than GPT-4o's 75.2%) while reducing cost by nearly 45%. On the MBPP code generation benchmark, COPE improves accuracy to 66.4% compared to GPT-4o's 64.0%, while cutting inference cost by nearly 75%. These results suggest that COPE is an effective framework for cost-aware LLM inference in the evaluated settings.

**Our contributions** are summarized as follows:

- We propose COPE, a test-time collaboration framework that brings planning into cost-aware inference through cross-model collaboration.

- COPE is training-free and few-shot-free, requiring only prompts at inference time, which makes it easy to deploy across tasks.

- Across diverse benchmarks, COPE substantially reduces monetary cost while maintaining performance comparable to large proprietary models.

- By interpreting confidence in a task-specific way, we present diverse applications of COPE across mathematical reasoning, code generation, open-ended generation, and sequential decision-making.

## 2 Related Work

**Planning for LLM inference.** Planning has been primarily studied in robotics and vision-language-action models as a core mechanism for structured decision-making (Kim et al., 2024; Black et al., 2023). A high-level planning module decomposes a task into subgoals, which are then carried out through low-level execution by downstream components. Planning can be seen as a feature of System-2 thinking (Ji et al., 2025).

In contrast, planning remains relatively underexplored in the context of LLM inference. Wang et al. (2023) propose the use of planning tokens prepended to each chain-of-thought step. However, this method requires supervised training to learn the tokens and focuses on local step-wise annotations rather than providing global guidance. Saha et al. (2025) introduce EvalPlanner, a framework for LLM-as-a-Judge (Zheng et al.,

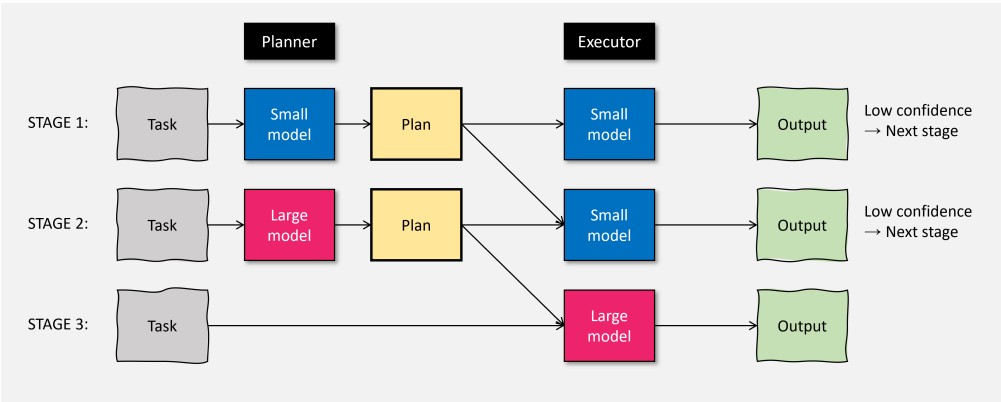

Figure 1: Overall framework of COPE. The system proceeds in up to three stages of inference, where small and large models alternate roles as planner and executor. In each stage, given a task, a plan is generated by the planner, and the executor produces candidate outputs. If a task-specific confidence falls below the stage threshold, the task escalates to the next stage. Plans generated in earlier stages are retained and reused in later stages.

2023) evaluation. Given a pair of responses, the model generates an evaluation plan to structure its judgment. However, the plan is generated only after seeing the responses and serves solely to structure the judgment process, not to guide generation. The method further requires training on both planning and reasoning.

In our method, planning precedes execution in LLM inference. The plan is a purpose-driven representation of the task (such as a goal or guideline). Here, planning is not learned and decoupled from execution, so it serves as a lightweight interface between models of different capacities.

**Test-time compute scaling.** Test-time compute scaling (Snell et al., 2024) refers to how computational resources are allocated and utilized at inference time to improve accuracy. These efforts typically fall into two directions: parallel sampling, which generates multiple outputs and selects the best among them (Wei et al., 2022; Wang et al., 2022), and sequential sampling (or iterative refinement), where a model revises its outputs over multiple passes (Madaan et al., 2023; Zelikman et al., 2022; Lee et al., 2025; Kumar et al., 2024; Muennighoff et al., 2025).

These techniques typically aim to improve performance by allocating more compute to a single model. However, they remain limited by the capacity of the model itself (Snell et al., 2024). This motivates a different approach: rather than pushing a single model further, we explore collaboration across models of different capacities. This form of cross-model scaling can be seen as a new axis of test-time compute scaling.

**Cost-aware LLM inference.** Some recent approaches aim to reduce the cost of LLM inference by using costly models only when necessary. Given a task, LLM Cascade (Yue et al., 2023) samples multiple answers from a free model leveraging task-specific few-shot exemplars of different formats, such as Chain of Thought (Kojima et al., 2022) and Program of Thought (Chen et al., 2022). It then estimates consensus ratio, which is the degree of agreement among the answers. If the consensus ratio fails to meet a criterion, the problem is escalated to a costly model. ABC (Kolawole et al., 2024) routes among more than two models without relying on curated few-shot examples. FrugalGPT (Chen et al., 2023) trains a lightweight scoring function for a routing policy. However, the limited capacity of the scoring function may hinder performance on complex tasks. In contrast, our method requires neither few-shot examples nor training.

## 3 Motivation

In this section, we provide motivating examples that explain why we structure small-large model collaboration through planning. Planning addresses a structural limitation of autoregressive generation and allows one

model to borrow high-level foresight from another via a low-cost plan. In this work, a *plan* is a high-level, purpose-oriented abstraction of the task that guides execution (refer to Figure 2).

Given a task, we first prompt the planner model to generate a brief guideline, typically one or two sentences, that outlines how to approach the task. This guideline is then provided, along with the original task, to the executor model. The following observations illustrate how this planning stage enables collaboration between models.

We use Llama-3.2-1B-Instruct, Llama-3.2-3B-Instruct, and GPT-4o-mini, hereafter referred to as Llama-1B, Llama-3B, and GPT-mini, respectively. Llama-1B and Llama-3B are open-source and freely accessible, while GPT-mini is accessed via a paid API.

**Observation 1.** *Larger planners help smaller executors.*

We begin by examining whether larger models can improve the performance of smaller ones by providing plans. As shown in Table 1, when a larger model acts as a planner for a smaller executor, such as GPT-mini planning for Llama-3B or Llama-3B planning for Llama-1B, the executor's accuracy improves over the no-plan baseline. For example, Llama-3B achieves 42.8% accuracy without planning, which rises to 53.0% when guided by GPT-mini. Likewise, Llama-1B improves from 25.2% to 36.4% when using plans from Llama-3B. This suggests that larger models are effective at guiding smaller models through planning.

**Observation 2.** *Smaller planners degrade larger executors.*

We next examine whether smaller models can effectively serve as planners for larger executors. As shown in Table 1, when smaller models such as Llama-1B or Llama-3B act as planners for larger models like GPT-mini or Llama-3B, the resulting performance is often lower than the no-plan baseline. For instance, GPT-mini achieves 73.8% without a plan, but drops to 70.6% and 69.6% when using plans from Llama-3B and Llama-1B, respectively. This suggests that low-quality plans generated by smaller models can hinder the execution ability of larger models.

**Observation 3.** *A model benefits from plans aligned with its capacity.*

We investigate whether a model can effectively act as their own planners, i.e., given a task, the model first generates a plan and then performs the task based on the plan. For large models such as GPT-mini, self-generated plans improve performance: GPT-mini achieves 73.8% accuracy without planning, which increases to 76.2% when using its own plans (Table 1). This suggests that large models can successfully scaffold their own execution through planning. In contrast, small models show mixed results when acting as their own planners. For example, Llama-1B performs worse with self-generated plans (23.2%) than with no plan at all (25.2%).

To explore whether simpler planning might help, we prompt the model to generate a goal (what to achieve) rather than a guideline (how to solve the task) as before. In this setting, accuracy improved to 30.2% (Table 2). This suggests that planning must be tailored to the model's capacity: small models benefit more from simpler purposes than from complex instructions.

Together, these observations suggest that planning can serve as a lightweight interface for cross-model collaboration, motivating our test-time framework COPE.

## 4 Method

We introduce COPE, a collaborative framework in which small and large models alternate roles as planners and executors across multiple stages. The core idea is to structure model collaboration by first generating a plan, which then guides downstream execution. Figure 1 illustrates the overall structure of COPE. Figure 2 shows an example of plan generation and execution based on that plan.

COPE operates in three stages. In Stage 1, a small model attempts to perform the task on its own by generating both a plan and an output. In Stage 2, a large model provides a new plan, and the small model reattempts execution with this guidance. In Stage 3, the large model assumes both roles to directly perform the task. Each stage is triggered only when the previous stage fails to reach sufficient confidence, enabling

---

**Vanilla**

**Problem:** You had 7 bags of gold coins, each containing the same number of coins. You found another bag that contained 53 coins, and you decided to redistribute all your coins evenly across the 8 bags. You successfully did this, and you also know that the total number of coins you have is more than 200. What is the smallest number of coins you could have had before finding the extra bag?
**Solution (executor):** Step 1: Determine the Target Number of Coins per Bag. Since you end up with eight bags and all bags have the same number of coins, let $x$ be the number of coins in each of the original seven bags. After adding the 53 coins, the total number of coins becomes $8x$. Thus, we have: $7x + 53 = 8x$ [...]

---

**COPE**

**Problem:** You have seven bags of gold coins. [...]
**Guideline (planner):** You want to redistribute these coins into 8 bags, meaning $7x + 53$ must be divisible by 8. Then, set up the equation $7x + 53 \geq 201$ to ensure you have more than 200 coins. Solve for $x$ to find the smallest integer that satisfies both conditions.
**Solution (executor):** Step 1: Set up the divisibility condition. The total coins $7x + 53$ must be divisible by 8. This can be expressed as: $7x + 53 \equiv 0 \pmod 8$. Simplifying modulo 8: $7x + 53 \equiv 7x + 5 \equiv 0 \pmod 8$, $7x \equiv -5 \pmod 8$. Since $-5 \equiv 3 \pmod 8$: $7x \equiv 3 \pmod 8$. [...] Step 3: Ensure $7x + 53 > 200$. Substitute $x = 8k + 5$ into the inequality: [...] The smallest integer $k$ that satisfies this inequality is $k = 3$. Therefore, the total number of coins before finding the bag of 53 coins is: $7x = 7 \times 29 = 203$. Thus, the smallest number of coins you could have had before finding the bag of 53 coins is: $\boxed{203}$

---

Figure 2: Comparison between vanilla inference and planning-guided inference with COPE. The vanilla model fails to account for the divisibility and inequality constraints, leading to incorrect reasoning (red). In contrast, COPE's planner highlights these constraints explicitly (green), allowing the executor to follow a structured solution path.

adaptive allocation of compute. Here, confidence is a general notion that can be instantiated in different ways depending on the task (e.g., majority voting for reasoning, test case pass rate for coding, perplexity for open-ended generation). Below we describe each stage in detail, focusing on majority voting as the confidence measure for reasoning tasks.

**Stage 1: small model planning and execution** In Stage 1, the small model acts as both planner and executor. Given a problem $x$, it first samples $n$ goal-type plans $g_1, \ldots, g_n \sim P_{\text{small}}(g \mid x)$. For each plan $g_i$, it then generates a solution $y_i \sim P_{\text{small}}(y \mid x, g_i)$, and extracts the corresponding answer $a_i = f(y_i)$, where $f$ denotes an answer extraction function.

The final answer is determined by majority vote over the answer set $\mathcal{A} := \{a_1, \ldots, a_n\}$, treating agreement among solutions as an indicator of correctness (Wang et al., 2022; Du et al., 2023). The *consensus ratio*, defined as the fraction of samples agreeing on the most frequent answer, serves as the confidence score:

$$\text{Consensus}(\mathcal{A}) := \frac{\max_{a' \in \mathcal{A}} |\{i : a_i = a'\}|}{n}. \tag{1}$$

If the consensus exceeds a threshold $\tau_1$, the answer is accepted. Otherwise, the problem is escalated to the next stage. In that case, one of the plans $g_i$ that produced the majority-voted answer is randomly selected and designated as $g^{\text{S}}$, which is passed to Stage 2.

**Stage 2: large model planning, small model execution** If Stage 1 fails to produce a confident answer, Stage 2 begins with the large model generating a new guideline-type plan $g^{\text{L}} \sim P_{\text{large}}(g \mid x)$. This plan is passed to the small model, which then attempts to solve the problem again by leveraging both the original plan $g^{\text{S}}$ from Stage 1 and the new plan $g^{\text{L}}$. These two plans are provided jointly as input to the small model in the prompt.

Table 3: Performance on the MATH-500 dataset for mathematical reasoning.

| Small | Large | Method | Shot | Train | Total Acc ↑ | Cost ↓ |
|-------|-------|--------|------|-------|-------------|--------|
| EXAONE-3.5 -2.4B | – | Small (single) | Zero | × | 52.8 | – |
| | – | Small (majority) | Zero | × | 65.2 | – |
| | GPT-4o | Large | Zero | × | 75.2 | 5025 |
| | | Cascade (vote) | Few | × | 72.4 | 3017 |
| | | Cascade (verify) | Few | × | 73.0 | 3021 |
| | | ABC | Zero | × | 74.4 | 4297 |
| | | FrugalGPT | Few | ✓ | 68.0 | 4163 |
| | | COPE (ours) | Zero | × | **75.8** | **2766** |
| | GPT-4o -mini | Large | Zero | × | 73.8 | 355 |
| | | Cascade (vote) | Few | × | 70.6 | 224 |
| | | Cascade (verify) | Few | × | 72.2 | 250 |
| | | ABC | Zero | × | 72.4 | 304 |
| | | FrugalGPT | Few | ✓ | 65.6 | 298 |
| | | COPE (ours) | Zero | × | **74.4** | **212** |
| Qwen-2.5 -1.5B | – | Small (single) | Zero | × | 49.0 | – |
| | – | Small (majority) | Zero | × | 58.0 | – |
| | GPT-4o | Large | Zero | × | 75.2 | 5025 |
| | | Cascade (vote) | Few | × | 71.4 | 5150 |
| | | Cascade (verify) | Few | × | 72.6 | 5046 |
| | | ABC | Zero | × | 72.6 | 4723 |
| | | FrugalGPT | Few | ✓ | 68.2 | 5472 |
| | | COPE (ours) | Zero | × | **75.2** | **3725** |
| | GPT-4o -mini | Large | Zero | × | 73.8 | 355 |
| | | Cascade (vote) | Few | × | 71.6 | 327 |
| | | Cascade (verify) | Few | × | 72.8 | 319 |
| | | ABC | Zero | × | 72.0 | 298 |
| | | FrugalGPT | Few | ✓ | 69.4 | 349 |
| | | COPE (ours) | Zero | × | **74.0** | **285** |

Then, the small model generates a solution $y_i \sim P_{\text{small}}(y \mid x, g^{\text{S}}, g^{\text{L}})$ and corresponding answer $a_i = f(y_i)$. Majority voting is applied over the answer set $\{a_1, \ldots, a_n\}$ as in Stage 1. If the consensus ratio exceeds a second threshold $\tau_2$, the answer is accepted. Otherwise, the problem is escalated to Stage 3.

**Stage 3: large model planning and execution**  If the small model still fails in Stage 2, the problem is escalated to Stage 3. Here, the large model takes full control of both planning and execution.

Specifically, it reuses the plan $g^{\text{L}}$ that it generated in Stage 2, and samples $y \sim P_{\text{large}}(y \mid x, g^{\text{L}})$. The final answer is extracted as $a = f(y)$. Since this is the final stage, no further escalation occurs, and the answer is returned as the system's output.

## 5   Experiments

We evaluate COPE on multiple tasks spanning mathematical reasoning, code generation, open-ended generation, and sequential decision-making. We find $n = 8$, $\tau_1 = 0.75$, and $\tau_2 = 0.5$ are effective in many of our experiments. Please refer to the empirical study on the effect of these hyperparameters in Appendix A.1. We summarize the sensitivity analyses from Appendix A.1. Increasing the sample count improves voting stability, while stricter consensus thresholds increase escalation and cost but reduce premature acceptance.

Table 4: Performance on the AIME-2024 dataset for mathematical reasoning.

| Method | Shot | Train | Acc ↑ | Cost ↓ |
|---|---|---|---|---|
| Small (single) | Zero | × | 3.33 | – |
| Small (majority) | Zero | × | 6.67 | – |
| Large | Zero | × | 33.3 | 502 |
| Cascade (vote) | Few | × | 23.3 | 568 |
| Cascade (verify) | Few | × | 26.6 | 570 |
| ABC | Zero | × | 33.3 | 485 |
| FrugalGPT | Few | ✓ | 33.3 | 513 |
| COPE (ours) | Zero | × | **40.0** | **478** |

**Baselines.** We compare COPE against several recent methods for cost-aware inference. We implement the method proposed in Yue et al. (2023), referred to as *Cascade*. Cascade includes two decision strategies: vote and verify. In the vote setting, multiple answers are sampled from two different few-shot prompts, and the final answer is selected by majority voting across all the answers. If the consensus ratio exceeds a threshold, the answer is accepted. Otherwise, the problem is escalated to a larger LLM. In the verify setting, majority voting is applied separately to answers from each few-shot prompt, and the final decision is accepted only if the two majority-voted answers agree. Otherwise, the problem is escalated to the larger LLM.

ABC (Kolawole et al., 2024) is another agreement-based method. Unlike Cascade, it does not rely on curated few-shot exemplars. Instead, it proceeds through multiple stages with progressively larger models as the stages advance. We report results using three models, following the original setup.

FrugalGPT (Chen et al., 2023) trains a lightweight scoring function which is implemented with Distil-BERT (Sanh et al., 2019). Specifically, the scoring function is trained to estimate whether an answer is correct for a given problem. A routing policy based on the score decides whether to accept the answer or escalate to a larger model.

We also consider three simpler baselines. The first uses only the small model in a single-pass setting without any sampling or aggregation (*Small (single)*). The second baseline uses the small model to generate multiple outputs and applies majority voting to select the most frequent answer (*Small (majority)*). The third baseline runs the large model once without planning or verification (*Large*). For each task, we evaluate against the baselines that are applicable to that task.

## 5.1 Mathematical reasoning

**Datasets and models.** We use the MATH-500 dataset (Lightman et al., 2023) and the AIME-2024 dataset (Art of Problem Solving, 2024). MATH-500 is a subset of the MATH dataset (Hendrycks et al., 2021) consisting of 500 math problems. The problems cover a broad range of topics and difficulty levels. Compared to MATH-500, AIME-2024 is a more challenging benchmark. AIME-2024 consists of 30 curated olympiad-level problems.

We use EXAONE-3.5-2.4B-Instruct (Research, 2024) and Qwen-2.5-1.5B-Instruct (Team, 2024) as the small models, and GPT-4o and GPT-4o-mini as the large models. For the more challenging AIME-2024 dataset, considering the increased difficulty of the problems, we use Qwen-2.5-3B-Instruct as the small model, and Gemini-2.0-Flash as the large model. For ABC, which requires three models, we additionally include Qwen-2.5-0.5B-Instruct alongside the above models.

**Evaluation metrics.** We evaluate accuracy (%) using exact match. For mathematical answers, we use symbolic comparison to accept equivalent expressions (e.g., 1/2 and 0.5). For non-mathematical outputs, exact string match is used. For cost, we use the actual per-problem API price based on real-world model pricing (OpenAI, 2025; Google DeepMind, 2024). We scale the values by one million for readability, reporting cost in USD per million problems.

Table 5: Performance on the MBPP dataset for code generation tasks.

| Small | Large | Method | Shot | Train | Total | |
|---|---|---|---|---|---|---|
| | | | | | Acc ↑ | Cost ↓ |
| EXAONE-3.5 -2.4B | – | Small (single) | Zero | ✗ | 42.2 | – |
| | – | Small (multi) | Zero | ✗ | 57.2 | – |
| | GPT-4o | Large | Zero | ✗ | 64.0 | 4889 |
| | | Cascade | Few | ✗ | 64.6 | 2706 |
| | | ABC | Zero | ✗ | 63.2 | 1267 |
| | | FrugalGPT | Few | ✓ | 63.6 | 4012 |
| | | COPE (ours) | Zero | ✗ | **66.4** | **1279** |
| | GPT-4o -mini | Large | Zero | ✗ | 57.6 | 276 |
| | | Cascade | Few | ✗ | 59.2 | 159 |
| | | ABC | Zero | ✗ | 59.4 | 80 |
| | | FrugalGPT | Few | ✓ | 53.4 | 240 |
| | | COPE (ours) | Zero | ✗ | **62.2** | **77** |
| Qwen-2.5 -1.5B | – | Small (single) | Zero | ✗ | 38.6 | – |
| | – | Small (multi) | Zero | ✗ | 54.2 | – |
| | GPT-4o | Large | Zero | ✗ | 64.0 | 4889 |
| | | Cascade | Few | ✗ | 62.2 | 2263 |
| | | ABC | Zero | ✗ | 62.6 | 1542 |
| | | FrugalGPT | Few | ✓ | 59.4 | 5029 |
| | | COPE (ours) | Zero | ✗ | **66.2** | **1469** |
| | GPT-4o -mini | Large | Zero | ✗ | 57.6 | 276 |
| | | Cascade | Few | ✗ | 57.4 | 147 |
| | | ABC | Zero | ✗ | 59.2 | 96 |
| | | FrugalGPT | Few | ✓ | 57.2 | 493 |
| | | COPE (ours) | Zero | ✗ | **61.4** | **94** |

**Results.** Table 3 shows that COPE matches the large model baseline in overall accuracy, with 75.8% vs. 75.2% for GPT-4o and 74.4% vs. 73.8% for GPT-4o-mini, while substantially reducing inference cost (2766 vs. 5025 for GPT-4o and 212 vs. 355 for GPT-4o-mini).

In Table 4, COPE achieves the highest mean accuracy while incurring the lowest cost among the compared methods. Because AIME-2024 contains only 30 problems, we interpret the accuracy difference cautiously and emphasize that COPE maintains competitive accuracy with reduced cost on this challenging benchmark. Compared to Table 3, the performance gap between the Small baselines and COPE is larger on these harder problems, which is consistent with the difficulty-based analysis in Appendix A.1.2.

## 5.2 Code generation

For code generation, we adopt a pass-based evaluation instead of majority voting, since multiple correct outputs may differ syntactically. Each output is executed against the task's test cases. If at least one output passes all test cases, a passing output is randomly selected as the final output. Otherwise, the task is escalated to the next stage.

**Datasets and models.** We use the 500-problem test set of the MBPP dataset (Austin et al., 2021), which consists of Python code generation tasks. Each task includes a task description and test cases. We use EXAONE-3.5-2.4B-Instruct and Qwen-2.5-1.5B-Instruct as the small models, and GPT-4o and GPT-4o-mini as the large models.

**Evaluation metrics.** We evaluate performance using the separate hidden test cases from the MBPP-ET dataset (Dong et al., 2025). A generated program is considered correct if it passes all hidden test cases.

Table 6: Performance on the MT-Bench dataset for open-ended tasks.

| Method | Win rate ↑ | Cost ↓ |
|---|---|---|
| Large | 42.5 | 11.06 |
| COPE (ours) | **44.8** | **8.64** |

Table 7: Performance on the ALFWorld benchmark for sequential decision-making.

| Method | Success rate (%) ↑ | Cost ↓ |
|---|---|---|
| Large | 35.0 | 225 |
| COPE (ours) | **36.9** | **160** |

**Results.** Table 5 reports the results on code generation. We find that, due to its limited capacity, the scoring function of FrugalGPT is difficult to train effectively on complex tasks. COPE achieves both the highest accuracy and the lowest cost. This shows that the benefits of collaborative planning extend beyond mathematical reasoning.

### 5.3 Open-ended tasks

To examine whether COPE extends beyond reasoning domains, we further evaluate it on open-ended tasks. In each stage, the small model generates multiple candidate outputs. The one with the lowest perplexity is selected as the final answer. If this perplexity is above a threshold, indicating low confidence, the task is escalated to the next stage.

**Datasets and models.** We use the MT-Bench dataset (Zheng et al., 2023), which consists of multi-turn open-ended prompts spanning domains such as writing and roleplay. We use EXAONE-3.5-2.4B-Instruct as the small model, GPT-4o-mini as the large model, and GPT-4o as the judge model.

**Evaluation metrics.** Since open-ended tasks lack unique ground-truth answers, we adopt the LLM-as-a-Judge protocol from MT-Bench (Zheng et al., 2023). A judge model compares two responses and assigns a win, loss, or tie. We report the win rate between COPE and the large model. We alternate the positions of outputs in pairwise comparisons to mitigate positional bias.

**Results.** Table 6 reports the win rate and inference cost on MT-Bench. COPE achieves a higher win rate than the large model baseline with much lower inference cost. Specifically, COPE obtains 36.9% wins, 28.1% ties, and 35.0% losses against the large model. This highlights that COPE can be instantiated on open-ended tasks. As with other LLM-as-a-judge evaluations, this result should be interpreted as a pairwise model-based preference evaluation rather than a substitute for human evaluation.

### 5.4 Agent tasks

We further evaluate COPE in an agent-task setting that requires multi-step decision-making. At every action step, the planner and executor collaborate to decide on next action, where the consensus is determined by majority voting.

**Datasets and models.** We conduct experiments on the evaluation split of the ALFWorld benchmark (Shridhar et al., 2020), which comprises 134 household tasks. In this benchmark, an LLM is employed as an agent, and it must predict actions over multiple steps by interacting with a text-based simulator. If the LLM-based agent successfully achieves the task, it receives the reward of 1.0; otherwise, 0.0. We use Qwen3-8B as the small model and GPT-4.1 as the large model.

**Evaluation metrics.** We report the success rate, which measures the percentage of tasks successfully resolved by the agent. In addition, we report the cost in USD, representing the total expense of running the evaluation on all 134 tasks.

**Results.** Table 7 shows that COPE can improve success rate while reducing cost in this agent-task setting.

Table 8: Planning ablations.

| Method | Planner | Executor | Verifier | Acc / Cost |
|---|---|---|---|---|
| *Full cascade* | | | | |
| No-plan cascade | – | Small/Large | Majority vote | 71.2 / 265 |
| COPE (ours) | Small/Large | Small/Large | Majority vote | 74.4 / 212 |
| *Single-stage collaborative* | | | | |
| Plan-only | Large | Small | Majority vote | 56.8 / 101 |
| Verify-only | – | Small | Large | 52.2 / 769 |
| *Large-only reference* | | | | |
| Self-plan | Large | Large | – | 75.6 / 435 |

## 5.5 Planning ablation

To isolate the contribution of planning from other components of COPE, we report full-cascade variants together with three targeted single-stage baselines. The experiments are conducted on MATH-500 using EXAONE-3.5-2.4B-Instruct as the small model and GPT-4o-mini as the large model.

We separate the ablations into two evaluation protocols. First, the full-cascade comparison preserves COPE's three-stage acceptance and escalation rule while removing only the generated plans. Second, the single-stage baselines compare alternative roles for the large model. Plan-only and Verify-only are single-stage collaborative alternatives. The former uses the large model before small-model execution as a planner, whereas the latter uses it after small-model generation as a verifier. Self-plan is a large-only reference that uses the large model for both planning and execution.

In Table 8, the full-cascade comparison first shows that adding plans improves accuracy while reducing cost relative to the no-plan cascade. Among the single-stage collaborative alternatives, using the large model as a planner is more effective than using it as a verifier. However, Plan-only remains far below COPE, indicating that a large-model plan alone is not sufficient, and it becomes effective when embedded in COPE's selective cascade. Finally, Self-plan serves as a large-model-only reference. It achieves comparable accuracy but requires about twice the cost of COPE. Overall, the results support the design choice of using planning as an intermediate interface within a selective cascade.

## 6 Limitations

COPE uses collaboration in an operational, workflow-level sense. Small and large models take complementary planner/executor roles and exchange intermediate plans. It is not intended to cover all possible forms of model collaboration. More tightly coupled interaction protocols, such as iterative critique or refinement between models, remain an interesting direction for future work.

COPE relies on task-dependent confidence signals, such as majority voting, unit-test pass rates, or perplexity. In domains where no reliable confidence signal is available, the cascade may accept incorrect small-model outputs or escalate too often. One can also consider alternative confidence estimators, such as lightweight reward models or self-verification prompts. However, reward models require additional supervision or training, increasing training cost, while self-verification prompts require extra LLM calls, increasing inference cost.

Finally, COPE shares a latency trade-off common to cascade-style routing methods. Because stages are executed sequentially, latency can increase compared with directly calling the large model, as shown in Table 15. COPE is therefore better suited to cost-sensitive or batch-style settings where API cost is the primary constraint, and may be less suitable for latency-sensitive interactive applications.

# 7 Conclusions

COPE is a test-time framework where small and large LLMs collaborate via planning. By exchanging low-overhead plans, it reduces inference cost while matching performance over strong baselines across diverse benchmarks. COPE aligns naturally with cost-sensitive edge-cloud settings, where many queries can be handled cheaply while larger models intervene selectively for harder cases.

**Broader Impact Statement**

As AI systems increasingly span both edge devices and the cloud, collaborative inference frameworks like COPE may help shape efficient and scalable deployment strategies. By reducing unnecessary compute, this approach can also contribute to more sustainable AI usage.

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

# A  Appendix

## A.1  Empirical Study

This section also includes sensitivity analyses for key hyperparameters, including the number of sampled solutions and the consensus thresholds used for routing. Unless otherwise noted, experiments in this section are conducted on the MATH-500 dataset using EXAONE-3.5-2.4B-Instruct as the small model and GPT-4o-mini as the large model.

Table 9: Truncated COPE variants.

| Setting | Accuracy (%) | Cost |
|---|---|---|
| Large | 73.8 | 355 |
| COPE (S1+S2+S3) | 74.4 | 212 |
| COPE (S1+S2) | 71.8 | 50 |
| COPE (S2+S3) | 73.4 | 248 |

Table 10: Effect of number of samples $n$ on accuracy and conditional accuracy under consensus.

| $n$ | Accuracy (%) | Accuracy @ Consensus (%) |
|---|---|---|
| 4 | 71.4 | 78.5 |
| 8 | 74.4 | 84.1 |
| 32 | 74.4 | 85.7 |

### A.1.1 Truncating COPE: balancing efficiency and accuracy

In Table 9, we compare a large model alone, full 3-stage COPE (S1+S2+S3), and two truncated variants: S1+S2 (early stop at Stage 2) and S2+S3 (skipping Stage 1).

The S1+S2 variant yields a dramatic reduction in inference cost ($212 \rightarrow 50$) with a moderate drop in accuracy ($74.4\% \rightarrow 71.8\%$). This suggests that, for many queries, once a stronger plan is introduced at Stage 2, further escalation is not always necessary.

The S2+S3 variant incurs higher cost, indicating that skipping the initial small-model stage reduces opportunities for early termination.

### A.1.2 Performance by difficulty

Table 11 shows the performance breakdown by difficulty level. Compared to the Small (majority) baseline, COPE achieves consistent improvements across all segments. On the easiest Level 1–2 problems, accuracy increases from 86.5% to 91.0% (+4.5). For intermediate Level 3–4 problems, the gain is larger, from 70.8% to 79.8% (+9.0). The improvement is most pronounced on the most challenging Level 5 problems, where accuracy rises from 34.3% to 49.3% (+15.0). These results indicate that the benefits of COPE become more significant as problem difficulty increases.

### A.1.3 Effect of sample count on voting stability

In Table 10, we study how the number of sampled solutions $n$ influences performance in our framework. Since the final answer is selected via majority voting under a consensus threshold, increasing $n$ does not always lead to higher overall accuracy. However, it introduces greater diversity in the reasoning paths explored, allowing the model to consider a broader range of plausible solutions.

Moreover, increasing $n$ improves the stability and reliability of voting outcomes. With more samples, the consensus ratio becomes less sensitive to sampling noise, and the decisions become more consistent. We observe that the conditional accuracy under consensus also increases with $n$, suggesting that when agreement is reached, it is more likely to reflect a correct answer. Overall, larger sample counts enhance both the breadth of exploration and the trustworthiness of decisions.

### A.1.4 High-consensus false positives

We analyze cases where the Stage-1 consensus exceeds the acceptance threshold but the accepted answer is incorrect. Table 12 reports the number of Stage-1 accepted queries and the false-positive rate among them. This analysis quantifies the risk of relying on consensus as a confidence proxy.

Table 11: Performance by difficulty. Improvements are larger for more difficult problems.

| Method | Level 1–2 | | Level 3–4 | | Level 5 | | Total | |
|---|---|---|---|---|---|---|---|---|
| | Acc | Cost | Acc | Cost | Acc | Cost | Acc ↑ | Cost ↓ |
| Small (single) | 78.9 | – | 54.9 | – | 23.1 | – | 52.8 | – |
| Small (majority) | 86.5 | – | 70.8 | – | 34.3 | – | 65.2 | – |
| COPE (ours) | 91.0 | 70 | 79.8 | 190 | 49.3 | 392 | 74.4 | 212 |

Table 12: High-consensus false positives.

| Stage-1 accepted | Incorrect accepted | False-positive rate (%) |
|---|---|---|
| 219.0 | 8.7 | 3.97 |

### A.1.5 Effect of consensus threshold

Table 13 shows how varying the consensus thresholds in Stage 1 and Stage 2 affects both accuracy and cost. When both thresholds are low (e.g., 0.5/0.5), the small model tends to accept answers even when the agreement among solutions is weak. This leads to lower cost, since fewer problems are escalated, but often results in reduced accuracy due to incorrect answers being accepted too early.

On the other hand, setting both thresholds high (e.g., 0.75/0.75) makes the system more cautious, escalating more problems to the large model. While this improves accuracy, it also increases inference cost. Among the tested combinations, the 0.75/0.5 setting offers a more favorable balance. This threshold configuration effectively reduces unnecessary use of the large model, without sacrificing much in terms of final accuracy.

### A.1.6 Cascading penalty

As a practical worst-case analysis, we examine queries that reach Stage 3, where COPE incurs the cost of earlier stages before invoking the large model directly. Table 14 reports the Stage-3 rate and the conditional cost penalty relative to directly calling the large model on the same subset of queries. Costs are averaged over queries that reach Stage 3. This shows the expected trade-off of adaptive cascading. Final-stage queries can incur extra cost, while the overall method reduces average API usage by resolving many queries earlier.

### A.1.7 Latency Tradeoff

Table 15 reports the average latency (sec) per problem measured on a single NVIDIA H100 GPU. As shown, the multi-stage structure leads to higher latency, but it also improves accuracy and significantly reduces cost. Since our primary focus is reducing inference API cost when using expensive cloud-based large models, this trade-off can be acceptable in scenarios where cost is the dominant constraint.

### A.1.8 Statistical uncertainty

To better assess small gains over the large-model baseline, we report mean performance, standard deviations, and 95% confidence intervals over five independent runs in Table 16. Overall, the results support the conclusion that COPE maintains comparable accuracy to the large-model baseline while consistently reducing paid API cost.

## A.2 Theoretical Interpretation

Each inference stage (Figure 3) follows three steps: (i) sample a *plan* $g$, (ii) generate a solution $y$ conditioned on $g$, and (iii) aggregate the extracted answers by majority vote. We formalize the first stage (the template for all stages) with a concise latent-variable model.

Table 13: Effect of consensus thresholds. Each row shows the number of problems resolved in each stage under different threshold settings for Stage 1 and Stage 2.

| Threshold | | Resolved Problems | | | Results | |
|---|---|---|---|---|---|---|
| Stage 1 | Stage 2 | Stage 1 | Stage 2 | Stage 3 | Accuracy (%) | Cost |
| 0.5 | 0.5 | 312 | 72 | 116 | 70.0 | 173 |
| 0.5 | 0.75 | 326 | 37 | 137 | 70.8 | 175 |
| 0.75 | 0.5 | 204 | 150 | 146 | 74.4 | 212 |
| 0.75 | 0.75 | 205 | 96 | 199 | 74.2 | 258 |

Table 14: Cascading penalty.

| Stage-3 rate (%) | Large-only cost | COPE cost | Penalty ratio |
|---|---|---|---|
| 28.7 | 501.8 | 624.8 | 1.25 |

**Latent variable model** We formalize prediction as computing the most likely answer $a = f(y)$, where $y$ is a solution and $x$ is the input problem:

$$\hat{a} = f\left(\arg\max_y P(y \mid x)\right),\tag{2}$$

where $f(\cdot)$ denotes an answer extraction function that maps a complete solution $y$ (e.g., a reasoning trace) to its final answer $a$ (e.g., a scalar result).

Although one could sample solutions directly from $P(y \mid x)$, we instead introduce a latent plan $g$ to structure the generation process. This yields the decomposition:

$$P(y \mid x) = \sum_g P(y \mid g, x) \cdot P(g \mid x).\tag{3}$$

This decomposition enables exploration of diverse reasoning strategies, each corresponding to a distinct plan. We sample from the resulting process by drawing $g_i \sim P(g \mid x)$ and then $y_i \sim P(y \mid g_i, x)$.

This process yields a set of solutions $y_1, \ldots, y_n$ sampled from $P(y \mid x)$. From each, we extract an answer $a_i = f(y_i)$. We then approximate the objective in Eq. equation 2 via majority voting:

$$f\left(\arg\max_y P(y \mid x)\right) \approx \text{mode}(f(y_1), \ldots, f(y_n)).\tag{4}$$

This procedure offers a simple approximation to the most likely answer under $P(y \mid x)$. Since higher-probability $y_i$ are sampled more often, their corresponding answers $f(y_i)$ dominate the vote. The latent plan $g$ encourages diversity by inducing varied reasoning strategies. This improves robustness, helping recover correct answers that might be missed by direct decoding.

**Bayesian inference** From a Bayesian perspective, we are interested in the posterior predictive distribution $P(a \mid x)$ over answers:

$$\sum_g \int_y \delta(f(y) = a) \, P(y \mid g, x) \, P(g \mid x) \, dy,\tag{5}$$

where $P(g \mid x)$ serves as a prior over plans and $P(y \mid g, x)$ is the likelihood of a solution path.

We approximate this distribution via Monte Carlo sampling: $g_i \sim P(g \mid x)$, $y_i \sim P(y \mid g_i, x)$, and set $a_i = f(y_i)$. Majority voting over $\{a_i\}$ yields an estimate $\hat{a} \approx \arg\max_a P(a \mid x)$.

We treat the consensus ratio as an empirical estimate of $P(a = \hat{a} \mid x)$, and use it to decide whether to accept the prediction or escalate.

Table 15: Latency trade-off of multi-round inference on a single NVIDIA H100 GPU (average per problem).

| Method | Acc ↑ | Cost ↓ | Latency ↓ |
|---|---|---|---|
| Large | 73.8 | 355 | 1.26 |
| COPE (ours) | 74.4 | 212 | 2.96 |

Table 16: Mean accuracy and cost over five runs.

| MATH-500 | | | | |
|---|---|---|---|---|
| Method | Acc ↑ | 95% CI | Cost ↓ | 95% CI |
| Large | $73.8 \pm 0.93$ | $[72.65, 74.95]$ | $355 \pm 3.4$ | $[350.78, 359.22]$ |
| COPE (ours) | $74.4 \pm 0.88$ | $[73.31, 75.49]$ | $207 \pm 6.0$ | $[199.55, 214.45]$ |
| AIME-2024 | | | | |
| Method | Acc ↑ | 95% CI | Cost ↓ | 95% CI |
| Large | $31.3 \pm 2.97$ | $[27.63, 35.02]$ | $498.7 \pm 6.6$ | $[490.49, 506.82]$ |
| COPE (ours) | $37.3 \pm 4.35$ | $[31.94, 42.73]$ | $478.2 \pm 4.8$ | $[472.23, 484.10]$ |

**Mixture-of-experts (MoE)**   Our approach shares several structural similarities with MoE models (Jordan & Jacobs, 1994; Cai et al., 2024):

First, sampling a plan $g \sim P(g \mid x)$ corresponds to selecting an expert based on the input, akin to input-dependent gating. Second, each plan defines a distinct strategy for solving the problem, much like how individual experts specialize in different regions of the input space. Third, given a selected plan, the model generates a solution via $P(y \mid g, x)$, which parallels how an expert produces an output conditioned on its assignment.

### A.3   Further Implementation Details

We conduct all experiments using 8 NVIDIA RTX-3090 GPUs. Our implementation builds upon and modifies the codebase from the `search-and-learn` repository by Hugging Face (Beeching et al., 2024). We provide our code in the supplementary material.

**Model version**   We use the default API versions of each large model and ensure consistency across all experiments. Specifically, GPT-4o-2024-08-06 is used for GPT-4o, GPT-4o-mini-2024-07-16 for GPT-4o-mini, and Gemini-2.0-Flash-001 for Gemini-2.0-Flash.

**Hyperparameters**   For code generation, we use a batch size of 32, and for mathematical reasoning, a batch size of 160. In COPE, decoding temperatures are set as follows: in Stage 1, the planner uses a temperature of 0.8 and the executor uses 0.4; in Stage 2, both the planner and executor use 0.8; and in Stage 3, the executor uses greedy decoding (temperature = 0).

**Prompts**   For mathematical reasoning, we adopt the system prompt from Beeching et al. (2024) for the executor. All other prompts, including those for planners are designed by us. The system prompts are provided in Appendix A.6.

### A.4   License

We use GPT-4o and GPT-4o-mini via their official APIs, which are subject to OpenAI's commercial license. LLaMA, EXAONE, Qwen models are used under Llama 3.2 Community License Agreement, EXAONE AI Model License Agreement, and Apache license 2.0, respectively. All datasets used in this work are publicly available and used under their respective licenses.

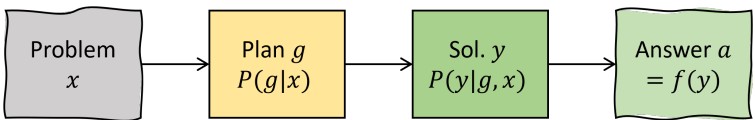

Figure 3: Common inference module in COPE.

## A.5 Qualitative Examples

---

**Vanilla**

**Problem:** Six cars pull up to a red light, one at a time. At the light, there are three lanes, one left-turn lane, one straight-going lane, and one right-turn lane. How many ways can the cars stack up so that all three lanes are occupied? Note that if the first car turns left and the second goes straight, this is considered different from the first car going straight and the second car turning left. In other words, the cars are distinguishable, but pull up to the intersection in a fixed order.

**Solution (executor):** To solve this problem, we need to determine the number of ways six distinguishable cars can be arranged such that each lane (left-turn, straight-going, right-turn) is occupied by exactly one car. Step-by-Step Solution: Step 1: Assigning Cars to Lanes Since there are three lanes and six cars, one lane will remain empty. We need to choose which lane will be empty. There are 3 ways to choose which lane will be empty (left-turn, straight-going, or right-turn). Step 2: Arranging Cars in Chosen Lanes Once we have chosen which lane will be empty, we need to arrange the remaining five cars in the three lanes. The number of ways to arrange $n$ distinguishable objects in $k$ distinguishable bins is given by $k^n$. Here, we have 5 cars to arrange in 3 lanes: $3^5 = 243$ Step 3: Combining Both Steps Since there are 3 ways to choose which lane will be empty and for each choice, there are $3^5$ ways to arrange the cars in the remaining lanes: $3 \times 243 = 729$ Therefore, the total number of ways the cars can stack up so that all three lanes are occupied is: $\boxed{729}$

---

**COPE**

**Problem:** Six cars pull up to a red light, one at a time. [...]

**Goal (planner):** The goal of the problem is to determine the number of distinct permutations of six distinguishable cars parking in three lanes (left turn, straight, right turn) such that each lane is occupied by at least one car.

**Guideline (planner):** To solve this problem, first determine the total number of ways to assign the cars to the three lanes such that each lane has at least one car. This can be achieved using the principle of inclusion-exclusion: calculate the total arrangements of cars in the lanes, subtract the arrangements where at least one lane is empty, and then add back arrangements where two lanes are empty. Finally, ensure that each arrangement respects the order of the cars.

**Solution (executor):** To solve this problem efficiently using combinatorial principles, we need to ensure that each lane (left-turn, straight-going, right-turn) has at least one car while accommodating six distinguishable cars in three lanes sequentially. Step-by-Step Solution: Step 1: Total Arrangements Without Restrictions. First, calculate the total number of ways to assign six cars into three lanes without any restrictions. Each car has 3 choices (left, straight, right): $3^6 = 729$. Step 2: Subtract Arrangements Where At Least One Lane Is Empty. Next, subtract the cases where at least one lane is empty. We use the principle of inclusion-exclusion for this purpose: One Lane Empty: Choose which lane is empty (3 ways), and distribute the cars among the remaining two lanes ($2^6$ ways for each choice): $\binom{3}{1} \cdot 2^6 = 3 \cdot 64 = 192$. Two Lanes Empty: Choose which two lanes are empty ($\binom{3}{2}$ ways), and place all cars into the remaining lane (1 way): $\binom{3}{2} \cdot 1^6 = 3 \cdot 1 = 3$. Using inclusion-exclusion: $729 - 192 + 3 = 540$. Step 3: Ensure Each Lane Has At Least One Car. Since we already ensured each lane had at least one car through subtraction in Step 2, no further adjustment is needed here. Conclusion. Therefore, the number of ways to stack up the six cars such that all three lanes are occupied is: $\boxed{540}$

---

Figure 4: Comparison between vanilla and COPE inference on a MATH-500 problem. The vanilla solution (left), generated by `EXAONE-3.5-2.4B-Instruct`, results in incorrect reasoning (red). In contrast, COPE combines a goal from the same model with a guideline from `GPT-4o-mini`, highlighting key constraints (green) and guiding the executor to a correct solution.

---

### Vanilla

**Problem:** Suppose $a$ and $b$ are different prime numbers greater than 2. How many whole-number divisors are there for the integer $a(2a + b) - 2a^2 + ab$?

**Solution (executor):** To determine the number of whole-number divisors of the integer $N = a(2a + b) - 2a^2 + ab$, where $a$ and $b$ are distinct prime numbers greater than 2, we start by simplifying $N$. Step 1: Simplify $N$. First, let's rewrite $N$: $N = a(2a + b) - 2a^2 + ab$. Expanding and combining like terms: $N = 2a^2 + ab - 2a^2 + ab = 2ab$. Thus, we have: $N = 2ab$. Step 2: Analyze the Divisors of $N$. Since $N = 2ab$ and $a$ and $b$ are distinct prime numbers greater than 2, we note the following: - $a$ and $b$ are both odd primes (since they are greater than 2). - The product $ab$ is odd because the product of two odd numbers is odd. - Therefore, $N = 2ab$ is an even number because it includes the factor 2. Step 3: Determine the Divisors. To find the divisors of $N = 2ab$: - The prime factorization of $N$ is $2^1 \times ab$. - The divisors of $N$ are all combinations of the factors $2^1$ and $ab$. The possible divisors are: $1, 2, ab, 2ab$. Conclusion. Counting these distinct divisors, we have: $\{1, 2, ab, 2ab\}$. Thus, there are **4** whole-number divisors of $N$. Therefore, the final answer is: $\boxed{4}$.

---

### COPE

**Problem:** Suppose $a$ and $b$ are different prime numbers greater than 2. [...]

**Goal (planner):** The goal of the problem is to determine the exact number of distinct whole-number divisors of the polynomial expression $a(2a + b) - 2a^2 + ab$, given that $a$ and $b$ are distinct prime numbers greater than 2.

**Solution (executor):** To determine the number of whole-number divisors of the expression $a(2a + b) - 2a^2 + ab$, we start by simplifying the expression algebraically. Step 1: Simplify the Expression. First, let's rewrite and simplify the given polynomial expression: $a(2a+b) - 2a^2 + ab$. Expanding and combining like terms: $a \cdot 2a + a \cdot b - 2a^2 + ab = 2a^2 + ab - 2a^2 + ab = 2ab$. Thus, the simplified form of the expression is: $2ab$. Step 2: Analyze Divisors of $2ab$. Since $a$ and $b$ are distinct prime numbers greater than 2, let's denote them as $a = p$ and $b = q$, where $p$ and $q$ are primes different from each other and greater than 2. The expression $2ab$ can be written as: $2pq$ Step 3: Determine the Divisors. To find the number of divisors of $2pq$, we use the property of divisors for a number $n = p_1^{e_1} p_2^{e_2} \cdots p_k^{e_k}$, which states that the total number of divisors is given by: $(e_1+1)(e_2+1) \cdots (e_k+1)$. For $2pq$: - The prime factorization is $2^1 \cdot p^1 \cdot q^1$. - Here, $e_1 = 1$ for 2, $e_2 = 1$ for $p$, and $e_3 = 1$ for $q$. Thus, the number of divisors is: $(1+1)(1+1)(1+1) = 2 \times 2 \times 2 = 8$. Conclusion. Therefore, the number of whole-number divisors of $a(2a + b) - 2a^2 + ab$ is $\boxed{8}$.

---

Figure 5: Comparison between vanilla and COPE inference on a MATH-500 problem. The vanilla solution (left), generated by `EXAONE-3.5-2.4B-Instruct`, results in incorrect reasoning (red). In contrast, COPE combines a goal from the same model, highlighting a key condition (green) and guiding the executor to a correct solution.

## A.6 Prompts

Table 17: System prompts used in COPE for mathematical reasoning

---

**Executor System Prompt**

Solve the following math problem efficiently and clearly:
- For simple problems (2 steps or fewer): Provide a concise solution with minimal explanation.
- For complex problems (3 steps or more): Use this step-by-step format:

```
## Step 1:  [Concise description]
[Brief explanation and calculations]

## Step 2:  [Concise description]
[Brief explanation and calculations]
```

Regardless of the approach, always conclude with:
Therefore, the final answer is: $\boxed{answer}$. I hope it is correct.
Where [answer] is just the final number or expression that solves the problem.

- - - - - - - - - - - - - - - - - - - - - - - - - - - - - - - - - - - - - - - - - - - - - - - - - - - - - - - - - - - - - - - - - - - - -

**Planner System Prompt**

**Goal Generation**: You are a math teacher. State the goal of the following math problem in one sentence.

**Plan Generation**: You are a skilled math coach. Explain how to solve the following math problem. Focus on strategy and key ideas. Respond in just one or two sentences. Return only the guidelines as plain text.

---

Table 18: System prompts used in COPE for code generation

---

**Executor System Prompt**

You are an expert Python programmer. Solve the following coding problem efficiently and clearly.

- - - - - - - - - - - - - - - - - - - - - - - - - - - - - - - - - - - - - - - - - - - - - - - - - - - - - - - - - - - - - - - - - - - - -

**Planner System Prompt**

**Goal Generation**: You are an expert Python programmer. State the goal of the following coding problem in one sentence.

**Plan Generation**:You are an expert Python programmer. Explain how to solve the following coding problem. Focus on strategy and key ideas. Respond in just one or two sentences. Return only the guidelines as plain text.

---

