# OpenReview forum: "Efficient LLM Collaboration via Planning"
_TMLR — Decision pending for TMLR_

### Review · Reviewer_He8Q · 2026-05-07

**Summary Of Contributions:**

This paper proposes COPE, a test-time collaboration framework for cost-aware LLM inference. The main idea is to use lightweight plans as an intermediate interface between small and large language models. The framework proceeds through multiple stages: a small model first plans and executes; if confidence is low, a large model provides a plan for the small executor; if confidence remains low, the large model performs the task directly.

The paper evaluates COPE on mathematical reasoning, code generation, open-ended generation, and agent tasks. The results suggest that planning-guided collaboration can reduce paid API usage while maintaining competitive performance.

The main strength of the paper is its practical motivation and simple training-free design. The main weakness is that the current evidence does not clearly isolate the contribution of planning itself from multi-sampling, routing, thresholding, and large-model assistance. Some reported gains over the large-model baseline are small, and parts of the experimental reporting are unclear.

**Additional Comments:**

This is a promising and practically motivated paper. However, the current version should be positioned more carefully as a planning-guided cascade, and the evidence needs to better separate the effect of planning from other components of the system. I would be open to reconsidering the work after these issues are addressed.

**Audience:**

Yes

**Audience Explanation:**

The paper studies a timely and practically important topic: how to combine small local models and larger API-based models for cost-aware inference. This should interest researchers working on efficient inference, test-time compute scaling, model routing, multi-model collaboration, and agent systems.

The idea of using plans as a lightweight interface between heterogeneous models is simple and potentially useful. Even if the current evidence is not yet sufficient for acceptance, the empirical direction is relevant to TMLR readers interested in practical LLM deployment and inference-time system design.

**Broader Impact Concerns:**

I do not have major broader impact concerns beyond those already discussed by the authors. The work mainly aims to reduce inference cost. However, the authors should note that cost-aware routing does not guarantee reliability, especially in settings where incorrect outputs may have high consequences.

**Claims And Evidence:**

No

**Claims Explanation:**

The evidence is promising but not fully convincing in its current form.

First, the paper claims that planning is the key mechanism enabling efficient small-large model collaboration. However, COPE combines several components: multi-sampling, confidence-based routing, cascade escalation, large-model assistance, and prompt-based planning. The current experiments do not sufficiently isolate the effect of planning from these other factors. Stronger compute-matched baselines are needed.

Second, some improvements over the large-model baseline are small. For example, on MATH-500, the reported gain over GPT-4o or GPT-4o-mini is around 0.6 percentage points in some settings, while the appendix reports overlapping standard deviations. The AIME-2024 result is also based on only 30 problems, so the apparent improvement may correspond to only a few additional correct answers.

Third, the MT-Bench and ALFWorld sections contain confusing metric descriptions. The MT-Bench text reports win/tie/loss statistics, while the table reports a “success rate.” The ALFWorld table is labeled as “win rate,” although the benchmark is normally evaluated by task success rate. These inconsistencies make the evidence harder to assess.

Finally, the cost analysis mainly accounts for paid API usage and treats local small-model inference as effectively free. This is reasonable for some deployment scenarios, but the paper also reports increased latency and uses local GPU inference. A fuller cost-latency-throughput analysis is needed to support the deployment claims.

**Requested Changes:**

1.The authors should add stronger baselines that isolate the contribution of planning, such as large-model plan + small-model execution without the full cascade, small-model sampling + large-model verification, and large-model self-planning under a comparable cost budget.

2.The authors should clarify and correct the MT-Bench and ALFWorld reporting. The current text and tables appear to mix win rate, success rate, and pairwise comparison results.

3.The authors should provide stronger statistical support for the main improvements, especially where the gains over the large-model baseline are small or based on small datasets.

---

> ### Author Response · Authors · 2026-06-07
>
> We deeply appreciate your insightful and constructive comments. They helped us clarify the scope of our claims, strengthen the analysis, and improve the presentation of the paper. Below, we respond to your comments one by one. In the revised draft, major revisions are marked in “blue”.
>
> ---
> ### **[R1]. "add stronger baselines that isolate the contribution of planning"**
>
> Our original baselines included competitive no-planning methods: Cascade can be viewed as a no-planning cascade with few-shot prompting, while ABC can be viewed as a no-planning cascade with an additional escalation stage. However, we agree that these baselines do not isolate the effect of planning cleanly. We therefore added a controlled no-plan cascade ablation in Section 5.5, which shows adding plans improves accuracy while reducing API cost.
>
> Following your suggestion, we also added single-stage collaborative alternatives. They show that using the large model as a planner is more effective than using it as a verifier. For Self-plan, we report the actual API costs since matching the budget on a per-query basis would require arbitrary choices. Self-plan achieves comparable accuracy but requires about twice the API cost of COPE.
>
>
> ---
> ### **[R2]. "clarify and correct the MT-Bench and ALFWorld reporting"**
>
> Thank you for pointing it out. The reported values were correct, but the MT-Bench and ALFWorld table headers were mislabeled. We have corrected them in the revision.
>
> ---
> ### **[R3]. "provide stronger statistical support"**
>
> We agree with your point. In Appendix A.1.8, we added an error-bar table for AIME-2024 and report 95\% confidence intervals for both MATH-500 and AIME-2024.
>
> For cost, the confidence intervals are clearly separated, supporting our main claim that COPE reduces paid API cost. For accuracy, although COPE has a higher mean, the confidence intervals overlap. We therefore revised the related claims to state the result more cautiously.

---

### Review · Reviewer_9mLv · 2026-05-20

**Summary Of Contributions:**

This paper proposes COPE (Collaborative Planning and Execution), a hierarchical inference framework in which a stronger LLM first generates high-level plans and weaker models attempt to execute them, with escalation to the stronger model only when confidence is low. The core idea is to use planning as a low-cost interface for transferring strategic guidance from large models to small models, thereby reducing inference cost while preserving much of the performance of stronger models. The framework is evaluated on mathematical reasoning, code generation, open-ended dialogue, and agent tasks. The paper reports that COPE achieves accuracy competitive with GPT-4-level models at substantially lower API cost, and argues that allocating compute to broader sampling is more effective than spending it on longer reasoning traces.

Key Strengths:
1. Plug-and-play simplicity: The framework is easy to deploy since it requires no finetuning, routing-model training, or carefully curated few-shot demonstrations.
2. Strong empirical cost-efficiency: The paper demonstrates substantial API-cost reductions across diverse tasks including mathematical reasoning, coding, and agent settings, while largely preserving strong-model performance.
3. Insightful cross-model scaffolding: The work provides convincing evidence that large models can effectively guide much weaker models through lightweight natural-language plans, which is a practically useful insight for edge–cloud deployment.

Key Weaknesses
1. Significant latency overhead is underemphasized: The multi-stage sequential sampling substantially increases inference latency (e.g., Table 12 reports an increase from 1.26s to 2.96s). While API cost decreases, the framework may be impractical for latency-sensitive interactive applications.
2. Overstated novelty and collaboration framing: The proposed “planning” mechanism often resembles automated strategic prompting or hierarchical prompting rather than genuine multi-agent collaboration.
3. Heavy reliance on heuristic hyperparameters: The framework depends on manually selected thresholds and sampling parameters, which may require retuning for new tasks and partially weakens the claimed plug-and-play simplicity.

**Audience:**

Yes

**Audience Explanation:**

The paper addresses an important practical question for modern LLM systems: whether strong models can efficiently guide weaker models through lightweight planning rather than full inference delegation. The proposed framework is simple, training-free, and broadly applicable across reasoning, coding, and agent tasks, making it relevant to researchers working on LLM systems, inference efficiency, cascaded routing, and multi-agent or hierarchical reasoning. In addition, the paper contains several useful empirical observations, particularly regarding planner quality, sampling versus extended reasoning, and the role of high-level strategic hints, that are likely to stimulate further discussion and follow-up work.

**Broader Impact Concerns:**

I do not see major direct ethical or societal risks arising from this work. The paper is primarily a systems and inference-efficiency study rather than a high-risk application domain.

**Claims And Evidence:**

No

**Claims Explanation:**

The paper provides interesting empirical evidence that high-level planning can improve the performance of weaker models and reduce API cost in several verifier-rich settings such as mathematics and coding. The qualitative examples are intuitive, and the ablations on planner quality, sampling, and routing provide some useful insights into hierarchical inference design. However, the evidence does not fully support several of the paper’s broader claims. In particular, the paper does not convincingly disentangle the gains from planning itself versus gains from increased test-time compute, repeated sampling, self-consistency, retries, and verifier-based filtering. The framing around “LLM collaboration” is also somewhat overstated, since the proposed mechanism often behaves more like automated strategic prompting or hierarchical prompting than genuine collaborative reasoning between models. In addition, the strongest empirical results occur in settings with external verification signals (e.g., unit tests or majority voting), leaving it unclear whether the framework generalizes to tasks without reliable confidence estimation. Finally, several conclusions about reasoning models and compute allocation appear stronger than what is strictly justified by the experimental scope and baselines. Overall, while the paper contains promising ideas and useful observations, the current evidence is not yet sufficiently rigorous or isolated to fully substantiate all of its central claims.

**Requested Changes:**

Critical:
1. The paper should include stronger controlled comparisons isolating the effect of planning itself from increased sampling, retries, self-consistency, and verifier-based filtering. Currently, it is unclear how much of the improvement truly comes from “collaboration” rather than simply allocating more inference-time compute.

2. The current presentation overstates the conceptual novelty of the framework. In many cases, the proposed mechanism behaves more like automated strategic prompting or hierarchical prompting than genuine collaborative reasoning between models. The paper should revise its framing accordingly and discuss this limitation more explicitly.

3. The strongest results are obtained in tasks with external verification signals such as unit tests or majority voting. The paper should explicitly discuss whether the framework remains effective in settings without reliable confidence estimation or external verifiers, and avoid overgeneralizing beyond the studied domains.

Strengthening:
1. A fairer evaluation would compare COPE against stronger single-model baselines using similar inference budgets (e.g., best-of-N sampling or self-consistency with the large model), rather than primarily against greedy single-pass decoding.

2. Since the framework substantially increases inference latency through sequential multi-stage execution, the paper should more clearly discuss which deployment settings are realistic for COPE and which are likely unsuitable due to latency constraints.

3. The framework appears sensitive to planner quality and manually chosen routing thresholds. Additional analysis of robustness across prompts, models, and hyperparameters would strengthen the credibility of the reported gains.

---

> ### Author Response · Authors · 2026-06-07
>
> We deeply appreciate your insightful and constructive comments. They helped us clarify the scope of our claims, strengthen the analysis, and improve the presentation of the paper. Below, we respond to your comments one by one. In the revised draft, major revisions are marked in “blue”.
>
> ---
> ### **[R1]. "include stronger controlled comparisons isolating the effect of planning"**
>
> We thank the reviewer for raising this important concern. We agree that the original evidence should more clearly separate planning from additional test-time compute. In the revision, we added a controlled planning ablation in Section 5.5. In the full-cascade comparison, we keep the same sampling, majority voting, acceptance thresholds, retries, and escalation rule, and remove only the generated plans. Under this controlled setting, COPE improves over the no-plan cascade in both accuracy and cost (74.4/212 vs. 71.2/265), suggesting that the gain is not explained by sampling or routing alone.
>
> This result is also consistent with the motivating planner-executor study in Tables 1 and 2, where changing the plan while holding the executor fixed substantially changes accuracy. Moreover, our original Cascade and ABC baselines are strong no-planning routing methods: they use sampling, agreement, and escalation, but do not pass plans between models. COPE's improvement over these baselines further suggests that planning provides benefits beyond simply allocating more inference-time compute.
>
> ---
> ### **[R2]. "revise its framing accordingly and discuss this limitation more explicitly"**
>
>
> We agree that a single COPE stage can resemble hierarchical prompting, since one model generates a plan and another conditions on it. However, our use of "collaboration" is at the workflow level: across the full cascade, small and large models take complementary planner/executor roles, pass intermediate plans across stages, and adaptively reallocate responsibility based on confidence. We revised the paper to clarify this operational meaning and to avoid overstating the conceptual framing.
>
> ---
> ### **[R3]. "discuss whether the framework remains effective in settings without reliable confidence estimation"**
>
> We agree with the broader concern that COPE depends on reliable confidence estimation, but we would like to clarify that not all confidence signals used in our experiments are external verifiers. Unit tests in code generation and environment feedback in ALFWorld are external signals, but majority voting/self-consistency in math is an internal agreement-based proxy, and perplexity in open-ended generation is a model-based confidence proxy. Thus, COPE does not require an external verifier in all settings; rather, it requires a task-specific confidence signal that is sufficiently correlated with output quality.
>
> We view this task-dependent confidence interface as a strength of COPE compared with methods tied primarily to majority voting. In our experiments, we instantiate the confidence signal differently across math, code, open-ended generation, and agent tasks. At the same time, we agree that COPE should not be overgeneralized to domains where no reliable confidence estimate is available. We added a Limitations section clarifying this dependence and noting that cost-aware routing should not be interpreted as a guarantee of correctness.
>
> ---
> ### **[R4]. "compare COPE against stronger single-model baselines using similar inference budgets"**
>
> We follow the standard evaluation in cost-aware routing work by comparing against single-call large-model baselines, which represent the direct paid-API deployment option. Large-model best-of-N or self-consistency baselines are informative but have a different cost structure: they spend paid API calls on every instance, whereas COPE adaptively invokes the large model only when earlier stages are uncertain. Exact per-instance budget matching would also be artificial because COPE's paid API usage is input-dependent. We therefore report large-only references under fixed protocols and compare their actual paid API costs.
>
> ---
> ### **[R5]. "discuss which deployment settings are realistic"**
>
> We appreciate this point. We now discuss the latency trade-off in the main Limitations section. Like other cascade-style routing methods, COPE can increase latency due to sequential multi-stage inference. We therefore clarify the intended deployment setting: COPE is most appropriate when reducing paid API usage is more important than minimizing end-to-end latency.
>
> ---
> ### **[R6]. "Additional analysis of robustness"**
>
> Regarding robustness, we note that the appendix includes analyses of key hyperparameters, including sample count (Table 10) and consensus thresholds (Table 13). To make this clearer, we now point readers to these analyses when discussing the robustness of COPE. The paper also evaluates COPE across multiple model combinations. We use simple task-level prompts throughout the experiments, rather than prompt-specific tuning.

---

### Review · Reviewer_UQrr · 2026-05-26

**Summary Of Contributions:**

The paper introduces COPE (Collaborative Planning and Execution), a training-free, test-time inference framework designed to balance the performance and cost of Large Language Models (LLMs). COPE operates as a multi-stage cascade where a smaller, less expensive model attempts to solve a task, and upon failing to reach a confidence threshold, escalates the task. Unlike traditional cascades that entirely hand off the task, COPE leverages the large model to generate a lightweight "plan" (e.g., a goal or guideline) to scaffold the smaller model's execution before resorting to a full large-model takeover. The authors evaluate COPE across mathematical reasoning, code generation, open-ended tasks, and agent tasks, reporting cost reductions alongside performance that is competitive with large proprietary models.

Key Strength:
- Ease of Integration: Because the framework is zero-shot and training-free, relying solely on prompt engineering, it is highly accessible for immediate deployment.
- Intuitive and Practical Concept: The paradigm shift from "complete handover" to "collaborative planning" is a highly practical approach to test-time compute allocation.

Key Weaknesses:
- Fragility of Consensus as a Fallback: The framework relies heavily on majority voting (consensus) to determine confidence. This assumes an LLM has a reliable, non-trivial understanding of its own capabilities, which is a flawed assumption when a small model suffers from systemic biases and confidently hallucinates incorrect answers.
- Cascading Cost Penalties: If the small model consistently fails, the multi-stage cascade actively increases inference costs and latency compared to a single large-model call, a vulnerability not thoroughly analyzed in the main text.
- Oversimplified Agentic Evaluation: The framework struggles to cleanly map onto modern, complex agentic systems. The evaluation relies solely on ALFWorld, which does not capture the dynamic, open-ended nature of modern multi-agent or tool-use architectures.

**Audience:**

Yes

**Audience Explanation:**

The TMLR audience includes many researchers and practitioners focused on test-time compute scaling, cost-aware inference, and edge-to-cloud deployment architectures. The core premise of COPE—using an expensive model to generate a cheap, lightweight plan rather than an expensive, full-length execution—is a highly relevant and interesting contribution to the current discourse on efficient LLM orchestration.

**Broader Impact Concerns:**

The authors provide a brief and adequate Broader Impact Statement regarding sustainable AI and scalable deployment. However, a minor addition is warranted regarding the reliability of the system in high-stakes domains (e.g., medical or legal reasoning). Because the system relies on small-model consensus to bypass large-model verification, systemic biases in the small model could lead to confidently incorrect automated decisions. The authors should briefly acknowledge the risks of deploying consensus-based cascades in critical real-world applications.

**Claims And Evidence:**

No

**Claims Explanation:**

While the empirical evidence on static benchmarks (e.g., MATH-500, MBPP) clearly supports the claim that COPE reduces cost while maintaining accuracy, the underlying claims regarding the robustness of the confidence mechanism and its applicability to agentic systems lack convincing evidence.

First, COPE relies on the consensus ratio to dictate escalation. The implicit claim is that consensus accurately reflects correctness. However, small models are notoriously prone to confident hallucinations; if a small model has a systemic misunderstanding of a prompt, it will repeatedly generate the same incorrect output, achieving high consensus and bypassing the escalation cascade entirely. The paper lacks a rigorous analysis of these false-positive consensus events.

Second, the paper claims COPE is effective for agent tasks, using ALFWorld as the sole benchmark. ALFWorld is a relatively constrained, text-based environment. Modern agentic systems involve complex tool use, multi-agent debates, and highly dynamic state changes. In these modern systems, passing high-level "guidelines" back and forth at every single atomic step introduces compounding errors and immense latency. The evidence provided is too narrow to support the broad claim that COPE is an effective prior for agentic settings.

**Requested Changes:**

1. Analysis of Confident Hallucinations: Provide a quantitative analysis of cases where the small model reaches a high consensus on an incorrect answer (false positives). Discuss the limitations of relying purely on majority voting for the escalation trigger, and how this impacts the reliability of Stage 1.

2. Worst-Case Cost Breakdown: Include a transparent analysis of the "cascading penalty." What percentage of queries on challenging datasets end up in Stage 3? For these specific queries, how much more expensive is COPE compared to simply calling the large model in the first place?

3. Refine Agentic Claims or Expand Evaluation: The ALFWorld evaluation is insufficient to claim broad applicability to modern agentic systems. Either evaluate COPE on a more complex, modern agentic benchmark (e.g., WebArena, SWE-bench) to prove it scales, or significantly tone down the claims regarding its effectiveness in agentic scenarios.

4. Alternative Confidence Metrics: Explore or discuss alternative, more robust fallbacks beyond simple majority voting, such as using a lightweight reward model or explicit self-verification prompts.

5. Latency Discussion: Expand on the latency trade-offs mentioned in the Appendix within the main text, as multi-round inference is a significant bottleneck in production environments.

---

> ### Author Response · Authors · 2026-06-07
>
> We deeply appreciate your insightful and constructive comments. They helped us clarify the scope of our claims, strengthen the analysis, and improve the presentation of the paper. Below, we respond to your comments one by one. In the revised draft, major revisions are marked in “blue”.
>
>
> ---
> ### **[R1]. Analysis of Confident Hallucinations**
>
> We added an analysis of high-consensus false positives in Section A.1.4. In our analysis, the false-positive rate among Stage-1 accepted queries is 3.97%. We also revised Section 6 to frame consensus as a task-dependent confidence proxy.
>
>
> ---
> ### **[R2]. Worst-Case Cost Breakdown**
>
> Following your suggestion, we added a Stage-3 cascading penalty analysis in Section A.1.6. The table compares the average COPE cost on queries that reach Stage 3 with the average large model cost on the same queries. Stage-3 queries incur a 1.25x cost relative to the large-only baseline. Stage-3 queries can be more expensive individually, while average API cost remains lower because many queries are resolved earlier.
>
>
> ---
> ### **[R3]. Refine Agentic Claims or Expand Evaluation**
>
> We appreciate this point and have clarified the scope of our agent evaluation. Our intention was to show that COPE can be applied to a sequential decision-making task, not to claim broad applicability to all agentic systems. Following the reviewer's suggestion, we carefully revised the agent-related wording throughout the paper to better reflect the scope of our evaluation.
>
>
> ---
> ### **[R4]. Alternative Confidence Metrics**
>
> Following your suggestion, we added a discussion in Section 6 on alternative confidence signals beyond majority voting, such as lightweight reward models and self-verification prompts, together with their training and inference-cost trade-offs.
>
> We also clarify that COPE is not tied to majority voting. In our experiments, we instantiate task-specific confidence signals beyond majority voting, such as unit-test pass rates for code and perplexity for open-ended generation.
>
> ---
> ### **[R5]. Latency Discussion**
>
> We appreciate this point. We now discuss the latency trade-off in Section 6. Like other cascade-style routing methods, COPE can increase latency due to sequential multi-stage inference.

---

### Review · Reviewer_t39Q · 2026-06-06

**Summary Of Contributions:**

COPE (Collaborative Planning and Execution) is a test-time inference framework where small and large LLMs alternate as planners and executors across a multi-stage cascade. The central idea is that a lightweight "plan," expressed as a goal or guideline, serves as a transferable interface between models of different capacities. The framework is training-free and few-shot-free, requiring only prompts at inference time.
The paper is grounded in three motivating observations: larger planners improve smaller executors, smaller planners degrade larger executors, and plan complexity must match executor capacity. COPE is evaluated across mathematical reasoning (MATH-500, AIME-2024), code generation (MBPP), open-ended generation (MT-Bench), and agent tasks (ALFWorld), reporting accuracy gains alongside 45-75% reductions in API inference cost. The work also reframes cross-model collaboration as a new axis of test-time compute scaling.

**Additional Comments:**

The paper is clearly written and well-motivated. Figure 2 comparing vanilla and planning-guided inference on a concrete math problem is an effective illustration of the mechanism. The related work section is thorough and accurately distinguishes COPE from EvalPlanner and planning token approaches. One clarification worth making explicit: the term "confidence" refers to majority vote ratio, test case pass rate, and perplexity depending on the task. While the paper acknowledges this, a more unified treatment in an appendix would improve reproducibility for practitioners extending COPE to new domains. Addressing the statistical rigor concerns and expanding the agent task analysis would bring this to a strong accept.

**Audience:**

Yes

**Audience Explanation:**

Cost-efficient LLM inference is among the most practically pressing problems in the field, and COPE's training-free, zero-shot design makes it accessible to a wide audience. Researchers working on test-time scaling, model routing, and collaborative AI systems will find the framework and its reframing of cross-model collaboration directly relevant. Practitioners deploying LLMs in edge-cloud hybrid settings will find the cost-accuracy analysis actionable. The insight that plan complexity must be calibrated to executor capacity is broadly useful beyond the specific framework proposed.

COPE sits at the intersection of several active research threads including test-time compute scaling, LLM cascades, and structured prompting, and offers a coherent alternative to existing delegation-only approaches. The empirical breadth across four task types and multiple model families gives the paper staying power as a reference point for future work. The motivating observations in Section 3, particularly around plan-capacity alignment, are insightful and carry implications for prompt design more broadly, independent of the COPE framework itself.

**Broader Impact Concerns:**

The primary impact of reducing inference cost is broadly positive and could support deployment in resource-constrained or privacy-sensitive settings where routing all queries to cloud models is undesirable. However, if the confidence signal is poorly calibrated, hard problems may be silently retained by the small model without user awareness, which is a reliability concern in high-stakes applications. The paper evaluates exclusively on English-language benchmarks, leaving generalizability to multilingual or low-resource settings unaddressed. There is also no discussion of how COPE behaves when the small and large models differ significantly in alignment or safety tuning, which is a realistic concern in practical deployments.

**Claims And Evidence:**

No

**Claims Explanation:**

1. The core performance and cost claims are supported by comparisons across multiple benchmarks and competitive baselines, and cost reporting is grounded in real-world API pricing, lending practical credibility to the efficiency claims.
The breadth of evaluation across four qualitatively distinct task types is a genuine strength, and the cost accounting methodology is more transparent than what is typical in the literature.
2. However, the AIME-2024 results derive from only 30 problems, meaning a reported 6.7 percentage point advantage corresponds to just two additional correct answers, which is too small a sample to support statistically reliable conclusions.
The open-ended task evaluation relies entirely on LLM-as-a-Judge via MT-Bench, introducing a potential circularity: GPT-4o serves as both the judge and the baseline large model, which may inflate win rates in COPE's favor due to shared stylistic preferences.
3. No statistical significance testing is provided anywhere in the paper. The absence of confidence intervals or variance estimates across all experiments makes it difficult to assess whether reported gains are robust or incidental.
4. Hyperparameter sensitivity analysis for consensus thresholds and sample count is deferred to an appendix without adequate coverage in the main text, yet these parameters directly govern escalation behavior and therefore the reliability of all reported results.
5. The agent task section reports only a single model pairing with no ablations, leaving its conclusions underexplored relative to the other task categories and making it difficult to attribute improvements specifically to the planning mechanism.
6. The paper does not analyze failure modes, particularly cases where a low-quality Stage 1 plan misleads the Stage 2 executor, a risk explicitly acknowledged in the motivating observations but never tracked systematically in the full system.

**Requested Changes:**

1. Statistical significance testing must be added across all main results. At minimum, confidence intervals or standard deviations over multiple runs are needed, particularly for AIME-2024 and ALFWorld given their small sample sizes.
The MT-Bench evaluation should address the judge-baseline overlap issue. Using an independent judge or supplementing with human evaluation would meaningfully strengthen the open-ended task claims.
2. The agent task section needs ablations examining how plan quality affects multi-step decision-making. A single model pairing with no analysis is insufficient to support the claims made.
3. Failure mode analysis is missing entirely. The paper should quantify how often a poor Stage 1 plan harms Stage 2 execution, given that this risk is acknowledged in the motivating observations.
4. Hyperparameter sensitivity for consensus thresholds and sample count should be summarized in the main paper, as these directly govern escalation behavior and therefore the reliability of all reported results.
5. Latency analysis should be included. For the edge deployment scenarios highlighted in the introduction, wall-clock latency across multi-stage inference may matter as much as API cost, and its absence is a conspicuous gap.
6. Plan quality should be evaluated independently to establish whether better plans causally drive better downstream performance, rather than leaving this relationship implicit.

---

> ### Author Response · Authors · 2026-06-07
>
> We deeply appreciate your insightful and constructive comments. They helped us clarify the scope of our claims, strengthen the analysis, and improve the presentation of the paper. Below, we respond to your comments one by one. In the revised draft, major revisions are marked in “blue”.
>
> ---
> ### **[R1]. Statistical significance testing**
>
> We expanded the statistical uncertainty analysis by reporting standard deviations and 95\% confidence intervals over multiple runs in Appendix A.2.8, and revised the related claims. For MT-Bench, both systems are evaluated by the same judge under the same protocol. We agree that LLM-as-a-judge is not a substitute for human evaluation. Accordingly, we revised the manuscript to describe the scope of the MT-Bench claim more cautiously.
>
> ---
> ### **[R2]. agent task section**
>
> The ALFWorld experiment is intended to show that COPE can be instantiated in a sequential decision-making setting. We therefore revised the manuscript to describe the scope of the ALFWorld claim more cautiously.
>
> ---
> ### **[R3]. Failure mode analysis**
>
> We added two failure-oriented analyses. First, the high-consensus false-positive analysis quantifies cases where Stage 1 confidently accepts an incorrect answer. Second, the Stage-3 cascading penalty analysis quantifies hard cases where the cascade reaches the final stage and incurs additional cost.
>
> ---
> ### **[R4]. Hyperparameter sensitivity**
>
> We clarify that the sensitivity analyses on consensus thresholds and sample count are included in the appendix. To make this clearer, we added a concise summary in the main text and explicitly point readers to the corresponding appendix tables.
>
> ---
> ### **[R5]. Latency analysis**
>
> We clarify that the latency analysis is included in the appendix, and we added a discussion of the latency trade-off to the main Limitations section.
>
> ---
> ### **[R6]. Independent role of plan quality**
>
> We clarify that the motivation experiments evaluate plan quality independently by changing the planner while holding the executor fixed. These results show that stronger planners improve downstream execution. We also analyze plan type by comparing goal-style and guideline-style plans, showing that plan complexity should match the executor's capacity.